# A Study of Assessment and Prediction of Water Quality Index Using Fuzzy Logic and ANN Models

Roman Trach, Yuliia Trach *, Agnieszka Kiersnowska, Anna Markiewicz, Marzena Lendo-Siwicka and Konstantin Rusakov

Institute of Civil Engineering, Warsaw University of Life Sciences, 02-776 Warsaw, Poland; roman_trach@sggw.edu.pl (R.T.); agnieszka_kiersnowska@sggw.edu.pl (A.K.); anna_markiewicz@sggw.edu.pl (A.M.); marzena_lendo_siwicka@sggw.edu.pl (M.L.-S.); konstantin_rusakov@sggw.edu.pl (K.R.)
* Correspondence: yuliia_trach@sggw.edu.pl; Tel.: +48-22-593-52-42

**Abstract:** Various human activities have been the main causes of surface water pollution. The uneven distribution of industrial enterprises in the territories of the main river basins of Ukraine do not always allow the real state of the water quality to be assessed. This article has three purposes: (1) the modification of the Ukrainian method for assessing the WQI, taking into account the level of negative impact of the most dangerous chemical elements, (2) the modeling of WQI assessment using fuzzy logic and (3) the creation of an artificial neural network model for the prediction of the WQI. The fuzzy logic model used four input variables and calculated one output variable (WQI). In the final stage of the study, six ANN models were analyzed, which differed from each other in various loss function optimizers and activation functions. The optimal results were shown using an ANN with the softmax activation function and Adam's loss function optimizer ($MAPE = 9.6\%$; $R^2 = 0.964$). A comparison of the $MAPE$ and $R^2$ indicators of the created ANN model with other models for assessing water quality showed that the level of agreement between the forecast and target data is satisfactory. The novelty of this study is in the proposal to modify the WQI assessment methodology which is used in Ukraine. At the same time, the phased and joint use of mathematical tools such as the fuzzy logic method and the ANN allow one to effectively evaluate and predict WQI values, respectively.

**Keywords:** water quality index; surface water; fuzzy logic; artificial neural network

## 1. Introduction

### 1.1. Water Quality Assessment

Surface water quality monitoring is a complex procedure that includes a number of chemical analyses of water. In Ukraine, nine river basins have been identified, and water quality monitoring is carried out according to nine chemical indicators. The areas of river basins are quite significant and range from tens of thousands to hundreds of thousands of square kilometers [1]. The quality of surface waters is influenced by human activities. The activities of industrial enterprises, the mining industry and others always affect the environment to a certain extent. Thus, the qualitative and quantitative analysis of surface waters is directly dependent on anthropogenic impacts. For example, if there is a deposit of heavy metals in the area of a river basin and it is extracted, then such heavy metals should be included in the list of controlled chemical indicators [2]. If a chemical industry facility is located in the area of a river basin, the specific chemicals produced by that facility should be included.

Given the uneven distribution of industrial enterprises in the territories of the river basins in Ukraine, monitoring the quality of surface waters by nine indicators does not allow the real state of the water near the river to be assessed. When analyzing the above, it

is expedient to select the necessary chemical indicators to monitor the quality of surface waters in accordance with a preliminary analysis of the anthropogenic impact on the basin of a particular river. Thus, it is advisable to make the list of controlled chemical indicators for water individually, depending on the type of anthropogenic activity [3,4]. The process of monitoring the quality of surface waters, including the systematic performance of chemical analyses of water, always comes with financial costs. If there are various industrial enterprises in the territory of a river basin, then the need to increase the number of controlled chemical indicators is obvious. In turn, this leads to an increase in monitoring costs. However, if there is no anthropogenic impact on the river basin, then the number of controlled chemical indicators of water quality can be reduced. This will reduce the cost of monitoring the water quality of such a river or an entire basin. It is important that with this approach, reducing the cost of monitoring such a river basin will increase the cost of monitoring a river basin with a high anthropogenic impact. There are various indices used to assess and monitor water quality in aquatic systems [5,6]. One of the first systems developed by Horton [7] was the creation of general indices that allow for the systematization of various water quality parameters. This methodology was then refined by the US National Sanitation Foundation (NSF), resulting in the well-known Water Quality Index (WQI) [8]. The WQI is an index that shows the level of cumulative influence of selected parameters on the overall water quality as a single numerical value [9,10]. This concept is widely used to assess water quality around the world [11,12].

A system for monitoring and assessing the quality of surface water, a method of examining individual sections of water in terms of their chemical, biological and nutritional components, has been introduced in many countries. Generic indices are used as comprehensive assessment tools that help assess water quality at an early stage and provide data and information for decision making by regulators. The assessment of water quality indicators makes it possible to establish the compliance or non-compliance of the water of a certain water body with requirements set by water users. The WQI has an advantage over other methods because it determines the overall water quality without interpreting individual factors [13]. Using a method that combines input parameters into a single resulting index has both advantages and limitations. The advantage is that the interpretation of the input variables is reduced to a single number, which makes it easier to understand the situation. The limitation of the method is due to the inability to assess individual factors, as well as the interdependence between them [14]. In addition, with numerous factors and data, calculating WQI can be time-consuming and difficult. Therefore, various mathematical models, including fuzzy logic and ANNs, deserve consideration as alternative tools for the assessment of water quality [15].

*1.2. Fuzzy Logic*

The use of mathematical modeling allows situations that arise and proceed in an uncertain environment to be simulated. Given the dynamic variability and a significant number of variables, there is a trend in the mathematical modeling of water quality to develop methods that minimize uncertainty and facilitate the numerical solution of problems. One of the methods is fuzzy logic, which generalizes classical set theory and formal logic. Fuzzy logic is an extension of classical logic and can be used to solve problems that have a significant amount of subjectivity. The use of fuzzy logic was first proposed by the scientist Lotfi Zadeh in 1965 [16]. The main reason for the appearance of a new theory was the presence of fuzzy reasoning in the description of processes, systems and objects by a person.

Fuzzy logic is capable of handling linguistic, vague and uncertain data and can be defined as a logical, reliable and transparent process of collecting and using data that creates opportunities for decision making in the environment. The uniqueness of fuzzy logic is that it allows complex environmental problems with numerous input variables and complex interdependencies between them to be solved [17,18]. Fuzzy logic tools and capabilities are used to assess water quality by calculating the WQI [19]. Modeling ecological systems

is a challenging scientific task because researchers often fail to make accurate statements about inputs and results. Fuzzy logic can be applied to the development of environmental monitoring indicators to solve this problem [20].

Due to its simplicity, fuzzy logic is successfully used to model natural-language-based water quality assessment [21]. Linguistic calculations used in fuzzy inference systems give better results than an algebraic expression for the estimation of the WQI. Fuzzy inference systems have been used to create water quality indices because these methods can provide alternative approximations when targets and boundaries are imprecise or poorly defined [22]. Thus, the authors conducted an extensive retrospective analysis of the evolution of methods for the calculation of the water quality index. Various options were analyzed, which concerned the choice of variables and the methods of weighting and aggregating these variables into a final value. The authors confirmed that the use of the fuzzy logic method can lead to significant progress in the methodology for the determination of the water quality index [23]. Caniani et al. [24] proposed to use a fuzzy model to assess the complex environmental vulnerability of an aquifer. The comparison of the obtained results with the traditional method showed that the fuzzy logic method turned out to be a useful and objective tool for environmental modeling. Yang et al. [25] created an early warning system aimed at accurately predicting algal blooms in rivers. The values of dissolved oxygen, velocity, ammonia nitrogen, total phosphorus and water temperature were used as input data for the fuzzy logic model. The fuzzy logic model successfully reproduced algal bloom events over a certain period of time. The authors of a study [26] used a Mamdani fuzzy logic model to classify groundwater quality for irrigation. The operation of the fuzzy model is based on the input membership functions of the electrical conductivity and sodium absorption coefficient, as well as on the output membership function of the irrigation water quality index.

Thus, the advantages of using fuzzy logic, compared to currently used water quality indices, are:

- The solution of issues with numerous input variables and complex interdependencies between them.
- The calculation of the final index occurs by evaluating the behavior of each analyzed parameter in relation to others.

### 1.3. Artificial Neural Network

A large amount of data have to be used to assess water quality. Traditional methods (for example, linear and non-linear regression) do not fully satisfy the needs of researchers, and artificial neural network (ANN) models come to the fore. ANNs are a family of models whose architecture is based on biological neural networks [27,28]. Scientists consider ANNs as a collection of artificial neurons that are systematized into one interconnected network. The neural network can detect implicit relationships between inputs and outputs and is able to predict the water quality index [29]. It is enough to train the network, and in the future, it will be able to predict values based on previous experience. In addition, ANN models are able to work effectively with a non-linear relationship between data and provide high accuracy of forecasts [30]. Creating an ANN requires an appropriate network structure, a number of inputs and outputs and the number of epochs used for simulation. The selection of the optimal network structure occurs by using experience and trial and error. Choosing the optimal network architecture, activation function, loss function and optimization algorithm is an important step to approximate complex non-linear relationships [31,32].

Modeling artificial neural networks for water quality prediction has been repeatedly used by scientists. Elkhatip and Komur [33] showed that the level of quality of forecasting by an ANN model has a strong dependence on the amount of initial data. Chen et al. [34] showed that ANN models demonstrate high potential for solving problems of the prediction of the quality of groundwater and surface water. Palani et al. [35] analyzed Multilayer Perceptron (MLP) and General Regression Neural Network (GRNN) models with various inputs chosen by incremental constructive methods for prediction. The authors proved

that a small dataset was a significant disadvantage for creating an optimal neural network. Wang et al. used a three-level MLP framework with a Back-Propagation (BP) algorithm to predict Chl-a levels. The dataset was divided into training (75%) and test (25%) samples. The results showed that an ANN model can effectively predict the value of the resulting indicator [36]. Miao et al. [37] used the Back-Propagation Neural Network (BPNN) to predict COD and ammonia nitrogen levels. A random non-linear relationship between input and output data was identified using the sigmoid function. Singh et al. [38] used eleven variables for the output layer. The data were split into three parts: 60% training set, 20% validation set and 20% testing set. As a result of using the neural network, the predicted output values were close to the real data.

Chen et al. [39] scaled the datasets so that the values were between 0 and 1, which allowed the use of a sigmoid transfer function. They applied constructive and clipping stepwise methods to maximize model performance by constantly adjusting predictions. Markus et al. [40] used trial and error to create an ANN architecture in their study. The result showed that the use of an ANN can improve the accuracy of $NO_3$ prediction compared to previous studies. Al-Mahallawi [41] argued that ANNs can model the complex process of water quality assessment because they provide a relationship between non-linear input and output data. Ai and Kisi [42] tested various ANN models. The results of the comparison showed that the Rotated Binary Neural Network (RBNN) model performs better than MLP in predicting the level of dissolved oxygen. Baek et al. [43] used modular neural networks (MNNs) that could effectively solve the problem of not sufficiently accurate prediction. They used momentum gradient descent and the back-propagation of the Levenberg–Marquardt error (TRAINLM). Chen and Liu [44] used a sigmoid function in the hidden layer and a linear function in the output layer. As a result, it was proven that the Adaptive Neuro-Fuzzy Inference System (ANFIS) and BPNN can predict DO with high accuracy. Han et al. [45] used cross-correlation for BOD prediction and cross-information for DO to select input data. Ta and Wei [46] applied the Adam optimization method which could handle sparse gradients on noisy issues to train the Convolutional Neural Network (CNN) parameters.

Thus, the novelty of this study is in the proposal to modify the WQI assessment methodology, which is used in Ukraine. At the same time, the phased and joint use of mathematical tools such as the fuzzy logic method and the ANN allow one to effectively evaluate and predict WQI values, respectively.

This article had three aims:

(1) To propose a diversified approach to assessing the quality of surface water using the number of analyzed chemical indicators depending on the degree of anthropogenic impact on a river basin;
(2) To model WQI evaluation using fuzzy logic;
(3) To create an artificial neural network model for WQI prediction.

The object of the study was to monitor data on the quality of water in the Western Bug River (Ukraine) for the period 2016–2021.

## 2. Study Area and Datasets

The assessment of the quality of surface waters is of great importance in the transboundary movement of hazardous pollutants along rivers from one state to another. This can contribute to negative changes in surface water and create threats to the environment and people. The Western Bug River flows from Podolsk Upland within western Ukraine (near Verkhobuzh, Lviv Oblast, Ukraine, coordinates 49°52′0.5736″ N 25°5′48.609″ E). At first, it flows in a westerly direction, but soon turns north. It flows past Brest (Belorussia) along the eastern outskirts of Lublin Upland (Poland) and further along Podlasie, flowing near Warsaw into the Narew River, not far from its confluence with the Vistula. The length of the river is 772 km, and the basin area is 73,470 km. The average slope is 0.8 m/km. The height of the source is 335 m above sea level. The height of the mouth is 68 m above sea level [47].

The Western Bug River is an unregulated river with a natural flow, but over the past 50 years, in the upper part of the basin, a number of its tributaries have been partially diverted, which has led to negative consequences in the ecological system. The surface waters of the Western Bug River are used to meet the needs for general types of water use, and are also used as a drinking water supply for settlements. The main sources of pollution in the Western Bug and its tributary are: wastewater discharges into the river without proper treatment; the unauthorized discharge of sewage; non-compliance with the regime in the coastal strips and water protection zones; coast erosion. Consequently, the anthropogenic factor has the greatest impact on the functioning of the river ecosystem, disrupting the natural state of the watercourse and introducing unusual components that degrade the water quality in the Western Bug River. The flow of pollutants with sewage into the Western Bug complicates the process of water treatment and requires an increase in energy costs for it. In this regard, establishing the causes, sources and extent of surface water pollution in this river and its tributaries is important, since even discharges of water treated according to the standard scheme into small rivers are accompanied by a sharp deterioration in water quality, posing a threat to public health. Monitoring the ecological state of surface waters of transboundary rivers is an important task in environmental activities, the result of which depend on efforts and funds from neighboring states.

For this study, the initial data (chemical indicators) were provided by the State Agency for Water Resources of Ukraine, which conducts systematic monitoring of the quality of surface waters. Data were collected from eight observation posts located in the Western Bug River (Figure 1). Observation posts were selected according to the criterion of the maximum matching of chemical indicators.

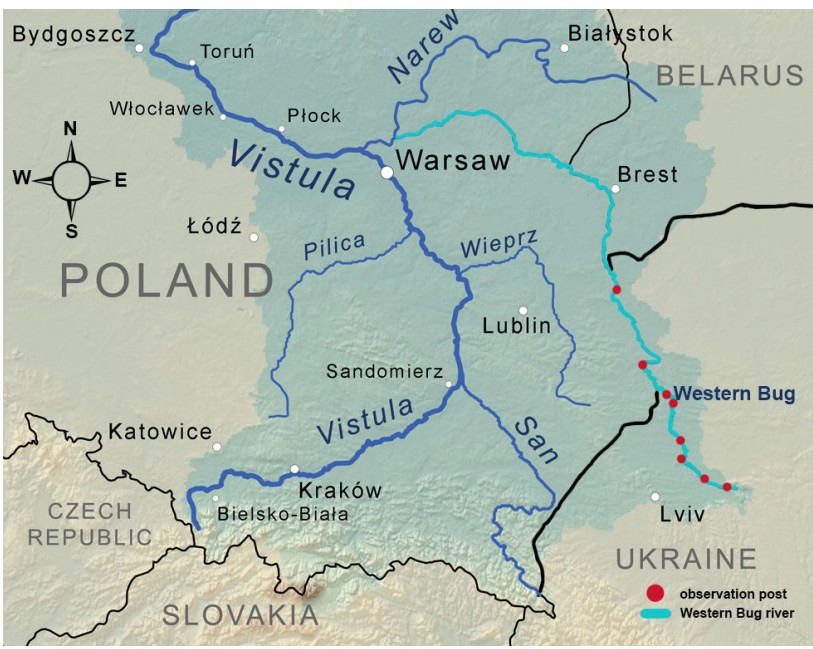

**Figure 1.** Observation posts in the territory of Ukraine in the Western Bug River.

In Ukraine, for the integral assessment of surface water quality, several methods were developed that take into account the mutual influence of certain indicators by calculating water pollution indices. The methodology that is used most often is the "Methodology for the ecological assessment of the quality of surface waters for the relevant categories", approved by the Ministry of Environmental Protection No. 89-M dated 4 June 2003 [48]. This methodology is based on the indicators of the chemical composition of water, and the criterion for assessing the admissibility of the content of substances in water is the multiplicity of exceeding the Maximum Contaminant Level (MCL) of harmful substances.

The assessment of water quality is carried out by calculating the Pollution Index (PI) (Formula (1)).

$$PI = \sum_{i=1}^{10} \left( \frac{1}{N_i} \sum_{n=1}^{N_i} x_i \right),$$ (1)

$$x_{in} = \begin{cases} if \ C_i > MCL \ then \ x_i = \frac{c_i}{MCL_i} \\ if \ C_i \leq MCL \ then \ x_i = 1 \end{cases}$$

where $i$—the index number, $N_i$—the total number of measurements of $i$ indicator, $x_i$—the multiplicity of MPL excess for $n$ measurement of $i$ indicator, $C_i$—the actual concentration of $i$ substance in water, $MCL_i$—the MPL $i$ of a substance in water.

Depending on the results of PI value, surface waters are divided into five quality categories: <1.00—good; [1.01, 2.50]—fairly good; [2.51, 5.00]—satisfactory; [5.01, 10.00]—bad; >10.00—very bad.

The output data were obtained from the website of the State Agency for Water Resources of Ukraine, Monitoring data (for the area of the river basin or sub-basin). The monitoring period was 1 January 2016–31 December 2021 to sub-basin–Vistula-Western Bug [49]. The number of chemical indicators was nine, the number of records in the database was 977. Table 1 shows an example of data from the Zabuzhye observation post for 22 August 2021.

**Table 1.** An example of chemical parameters obtained at the Zabuzhye post.

| Indicator | Actual Value | MCL |
|---|---|---|
| Biochemical oxygen consumption in 5 days (BOD$_5$), mgO/dm$^3$ | 2.3 | 3 |
| Suspended substances (SS), mg/dm$^3$ | 22 | 15 |
| Dissolved oxygen (DO), mgO$_2$/dm$^3$ | 8.5 | 4 |
| Sulfate ions (SO$_4^{2-}$), mg/dm$^3$ | 52 | 100 |
| Chloride ions (Cl$^-$), mg/dm$^3$ | 19 | 300 |
| Ammonium ions (NH$_3$), mg/dm$^3$ | 0.18 | 0.5 |
| Nitrate ions (NO$_3^-$), mg/dm$^3$ | 0.11 | 40 |
| Nitrite ions (NO$_2^-$), mg/dm$^3$ | 0.016 | 0.08 |
| Phosphate ions (PO$_4^{3-}$), mg/dm$^3$ | 0.046 | - |

## 3. Methods

### 3.1. Fuzzy Logic Modeling

The next stage of the study was the modeling of the WQI assessment using fuzzy logic. After analyzing the methodology of a number of previous studies in which the fuzzy logic methodology was used to assess the water quality indicator, it can be argued that the main difference was in the number of input variables used and, accordingly, the number of rules created. For example, Raman et al. [50] used 6 variables and created 86 rules. Lermontov et al. [22] used 9 variables and created 3125 rules. Semiromi et al. [51] used 6 variables and created 58 rules. Gharibi et al. [21] used 20 variables and created 550 rules. Tiri et al. [52] used 10 variables; however, the number of rules was not specified.

All of these studies used the Mamdani-type fuzzy inference system. Fuzzy logic is a process of decision making and the evaluation of situations by an expert in the form of an algorithm consisting of three main stages: fuzzification, aggregation and defuzzification. In this study, the fuzzy inference system was also implemented using Mamdani-type inference, which is the most commonly used method and is based on a set of "If-Then" rules.

At the initial stage (the fuzzification stage), a set of clear input data was collected, which, using fuzzy linguistic variables, fuzzy linguistic terms and membership functions, was transformed into a fuzzy set.

The characteristic of the fuzzy set is the membership function. In this study, a triangular membership function was used, which can be expressed as:

$$\mu_\Delta(x) = \begin{cases} 0, & x \leq x_1 \\ \frac{x-x_1}{x_2-x_1}, & x_1 \leq x \leq x_2 \\ \frac{x_3-x}{x_3-x_2}, & x_2 \leq x \leq x_3 \\ 0, & x > x_3 \end{cases} \tag{2}$$

where $x_1, x_2, x_3$ are numeric parameters that can take arbitrary values and are ordered by the ratio $x_1 < x_2 < x_3$.

Each of the four input variables corresponds to a triangular membership function with three ranges of values that correspond to three levels: low, medium and high [53]. The next stage (aggregation stage) includes a rule base and an inference algorithm based on the membership function. The rule base contains logical causal relationships between input and output variables. Fuzzy aggregation is central to fuzzy logic. This system connects the basic concepts of fuzzy logic: membership functions, linguistic variables and fuzzy logical operations [54]. At the defuzzification stage, based on the created rule base and using membership functions, the resulting fuzzy inference was transformed into a crisp inference [55].

### 3.2. ANN Modelling

The final stage of the study was to create an artificial neural network model for WQI prediction. In this study, six ANN models were created, and their effectiveness was analyzed. The preparation and normalization of input variables, as well as the modeling, training and validation of the ANN, were carried out using the Keras library in Python language. Visualization was implemented using the Matplotlib library in Python language.

The ANN architecture shown in Figure 2 has:

— The input layer, which consists of four variables;
— The first hidden layer, which has 64 neurons;
— The second hidden layer, which has 32 neurons;
— The output layer, which consists of a single output variable.

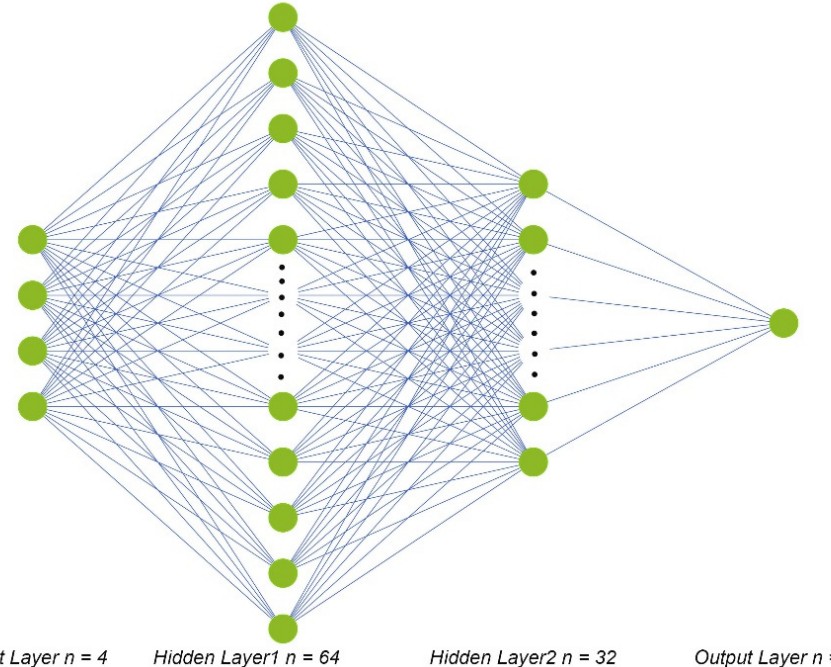

*Input Layer n = 4    Hidden Layer1 n = 64    Hidden Layer2 n = 32    Output Layer n = 1*

**Figure 2.** Scheme of ANN architecture.

The number of hidden layers and neurons in them was determined by trial and error.

ANN models were analyzed using three activation functions: sigmoid, softmax and ReLU.

The sigmoid function is defined as [34]:

$$f(x) = \frac{1}{1 + e^{-x}} \tag{3}$$

Using the sigmoid function in a network with a large number of neurons can lead to the activation of almost all neurons. In turn, this will reduce the performance of the model. In such a situation, the ReLU function has the advantage. Using ReLU allows one to activate fewer neutrons, which improves the performance of the entire network. If the argument $x < 0$, then the function $f(x)$ is equal to 0; if $x \geq 0$, then the function $f(x)$ returns the number itself:

$$f(x) = \begin{cases} 0 \ for \ x < 0 \\ x \ for \ x \geq 0 \end{cases} \tag{4}$$

Softmax is a multivariate generalization of the logistic function and converts a vector of numbers into a vector of probabilities. Softmax is used to normalize the output by converting it from weighted sum values to probabilities that amount to 1. The softmax function transforms a vector $z$ of dimension $K$ into a vector $\sigma$ of the same dimension, where each coordinate $\sigma_i$ of the resulting vector is represented by a number in the interval [0, 1] and the total of the coordinates is 1.

The coordinates $\sigma_i$ are calculated as follows:

$$\sigma(z)_i = \frac{e^{z_i}}{\sum_{k=1}^{K} e^{z_k}}. \tag{5}$$

The next important step in ANN modeling is the selection of loss function. The loss function calculates the difference between the actual and target values for each neuron, thereby estimating the accuracy of the prediction. The network is trained until the global minimum error is reached.

To calculate the loss function, the root mean square error (*MSE*) was used:

$$MSE = \frac{1}{n} \sum_{i=1}^{n} (Y'_i - Y_i)^2. \tag{6}$$

where $Y'_i$ is the output calculated by the model, and $Y_i$ is the target output.

Each dataset (difference between actual and target values) was squared, and then the values were summed and divided by the total number of datasets. To eliminate overfitting, the early stopping method was used [56]. Network training stopped when the monitored metric (loss function) no longer showed improvement. The learning cycle checked to see if the loss decreased at the end of each epoch. Various optimization algorithms were used to minimize the loss function. We tested two methods: stochastic gradient descent (SGD) and Adaptive Moment Estimation (Adam). The algorithm of SGD uses one training set at each step and updates the weights of neural network [57]. Adam's method is efficient for significant calculations, requires little memory and is well suited for problems with large amounts of data and parameters [58]. The quality metric mean absolute error *(MAE)* was used to evaluate the performance of the models. The *MAE* indicator is determined using Formula (7) [59]:

$$MAE = \frac{1}{n} \sum_{i=1}^{n} |Y'_i - Y_i|. \tag{7}$$

A comparative analysis of the performance of six neural networks was carried out using the mean absolute percentage error *(MAPE)* and the coefficient of determination ($R^2$).

The *MAPE* indicator is determined using Formula (8) [60]:

$$MAPE = \frac{100\%}{n} \sum_{i=1}^{n} \left| \frac{Y_i - Y_i'}{Y_i} \right|. \tag{8}$$

$R^2$ is a statistical measure used to predict future outcomes or test hypotheses based on other related information. $R^2$ is determined using Formula (9) [61]:

$$R^2 = 1 - \frac{\sum (Y_i - Y_i')^2}{\sum (Y_i - \overline{Y_i})^2}. \tag{9}$$

where $\overline{Y_i}$—the mean of the target output data.

## 4. Results and Discussion

In the first stage of the research, the collection, systematization and processing of the available initial chemical information on the quality of water in the Western Bug River was carried out.

As mentioned earlier, the methodology for assessing the quality of surface waters in Ukraine is based on the use of nine chemical indicators (Table 1), and it does not differentiate chemical indicators in terms of the level of negative impact. The analysis of these indicators and the level of their influence on the functioning of flora and fauna showed that they can be divided into three groups.

Group I ($O_2$ and $BOD_5$) indicators are the most important and require constant monitoring. The concentration of soluble oxygen in water over 4 $mgO_2/dm^3$ can accelerate the processes of the oxidation of organic substances to $CO_2$ and $H_2O$ [62]. It should be noted that with a decrease in the concentration of soluble oxygen, the pH value of water may decrease while its acidity may increase. There are almost always heavy metals and phosphates at the bottom of every body of water in immobile form. Increasing the acidity of water can cause the dissolution of mineralized forms of heavy metals and phosphates, thereby seriously deteriorating the condition of water.

Group II ($NO_3^-$, $NO_2^-$, $NH_4^+$)—chemical indicators from this group may have a lesser negative impact compared to group I if the $O_2$ concentration is less than 4 $mgO_2/dm^3$. The concentrations of $NO_2^-$ and $NH_3$ indicators will not have high values since they will be oxidized to $NO_3^-$. In addition, this value of $O_2$ concentration will make it possible to maintain the pH of water within the range of a neutral and slightly alkaline environment, and, consequently, the newly formed $NO_3^-$ will not reduce the pH of water. At these pH values, the $NH_3$ present will be less toxic than in an acidic environment [62].

Group III ($SO_4^{2-}$, Cl, $PO_4^{3-}$, SS)—chemical indicators from this group are present in water due to the release of highly concentrated wastewater and mine water from the mining industry. Therefore, when analyzing the anthropogenic impact on a water body and the absence of such sources of pollution, monitoring for $SO_4^{2-}$ and $Cl^-$ can be omitted [63,64]. The presence of phosphates in high concentrations in water is associated with anthropogenic impacts [65]. Elevated phosphate concentrations cause the phenomenon of eutrophication and, consequently, an increase in suspended solids. Thus, we can say that the increase in suspended solids depends on the increase in phosphates in water.

In this regard, we propose to divide the water quality assessment process into two stages (Figure 3).

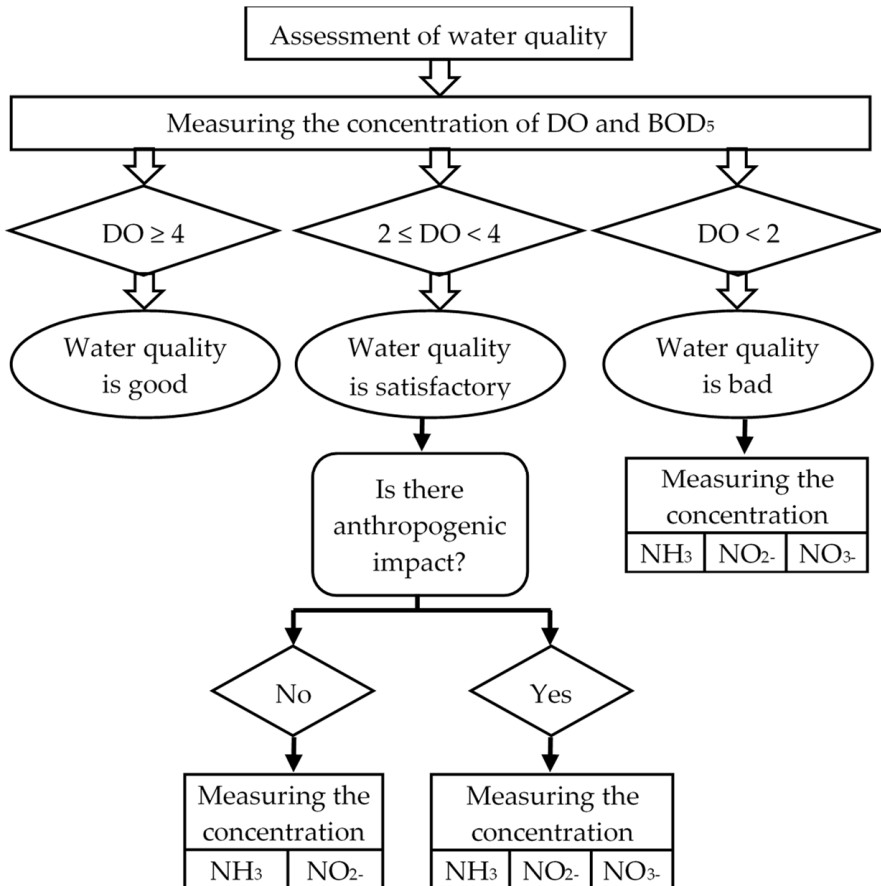

**Figure 3.** Step by step water quality assessment process.

Stage 1 consists of measuring the concentration of DO and $BOD_5$. The level of DO value determines which chemical indicators will be measured in stage 2. Thus, if $DO \geq 4$ $mgO_2/dm^3$, then the water has the highest quality level "good". If the value of DO concentration is in the range from 2 to 4 $mgO_2/dm^3$, it is necessary to additionally measure the concentrations of $NH_3$ and $NO_2^-$. The water body will have anoxic living conditions for flora and fauna. If the value of DO concentration $< 2$ $mgO_2/dm^3$, then it is necessary to measure both $BOD_5$ and all forms of nitrogen $NH_3$, $NO_2$, $NO_3$. The water body will have anaerobic living conditions for flora and fauna.

Four chemical indicators were used as input variables: $BOD_5$, $NH_3$, $NO_2$ and $NO_3$. The fuzzy logic system was created and tested using the Fuzzy logic Toolbox Matlab R2015b. The triangular membership function was used for the fuzzification of the input variables (Figure 4). Each of the variables was described by three terms: low, medium and high. The linguistic value of the $BOD_5$ indicator is in the ranges: low—[−6, 0, 6]; medium—[1.5, 7.5, 13.5]; high—[9, 15, 21]; $NH_3$ indicator is in the ranges: low—[−1.2, 0, 1.2]; medium—[0.3, 1.5, 2.7]; high—[1.8, 3, 4.2]; $NO_2$ indicator is in the ranges: low—[−0.16, 0, 0.16]; medium—[0.04, 0.2, 0.36]; high—[0.2, 0.4, 0.56]; $NO_3$ indicator is in the ranges: low—[−32, 0, 32]; medium—[8, 40, 72]; high—[48, 80, 112]. To calculate the output value, the centroid defuzzification method was used, in which the output value was determined based on the center of the weight of the output fuzzy set.

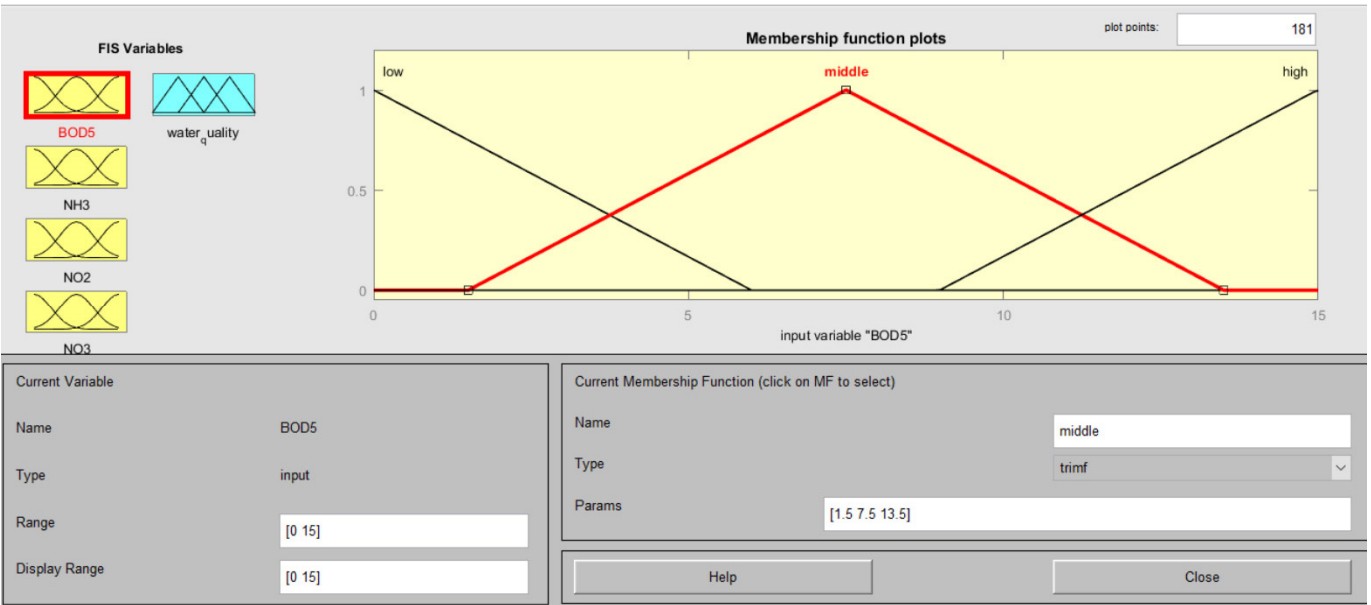

**Figure 4.** An example of a triangular membership function (BOD$_5$ indicator).

The inference system checks the values of each linguistic variable using fuzzy logic rules and transforms the input set into an output linguistic variable. The next step in fuzzy inference is to aggregate the output data based on the generated rules. Numerous rules are simultaneously processed with their further aggregation into the final solution with fuzzy inference. The set of rules in this study included 25 rules (Figure 5). This study used a Mamdani-type inference based on the logical function "If-Then", using the "And" function as a connector. Thus, If the values of the input indicators corresponded to the level of BOD$_5$—"low" And NH$_3$—"middle" And NO$_2$—"middle" And NO$_3$—"middle", Then the output value WQI was equal to "fairly bad".

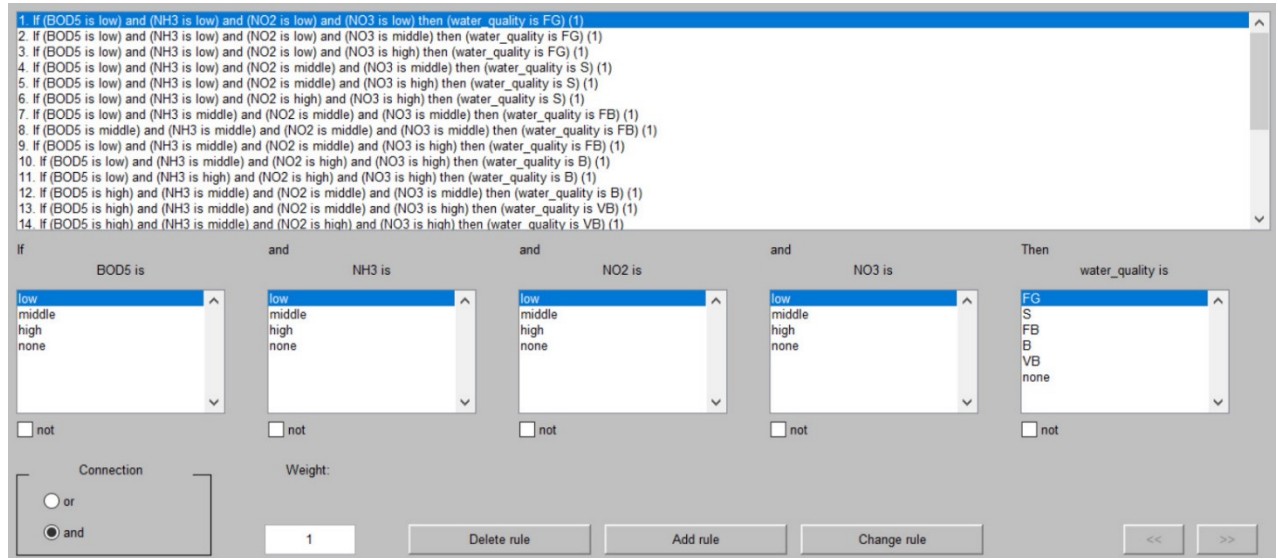

**Figure 5.** Fuzzy rules.

An example of the rules used in this study is shown in Table 2. Thus, if the values of the input indicators corresponded to the level of "low", "middle", "middle", "middle", then the output value was equal to "fairly bad".

**Table 2.** Example of rules used in the study.

| BOD$_5$ | NH$_3$ | NO$_2$ | NO$_3$ | WQI |
|---|---|---|---|---|
| **low** | low | middle | middle | fairly good |
| **low** | low | middle | high | satisfactory |
| **low** | middle | middle | middle | fairly bad |
| **high** | high | middle | low | bad |
| **high** | middle | middle | high | very bad |

The linguistic output variable WQI could take one of five values: fairly good (FG), satisfactory (S), fairly bad (FB), bad (B) or very bad (VB) (Figure 6). The linguistic value of the BOD$_5$ indicator was in the ranges: fairly good—[−1.5, 0, 1.5]; satisfactory—[1, 2.5, 4]; fairly bad—[3.5, 5, 6.5]; bad—[6, 7.5, 9]; very bad—[8.5, 10, 11.5].

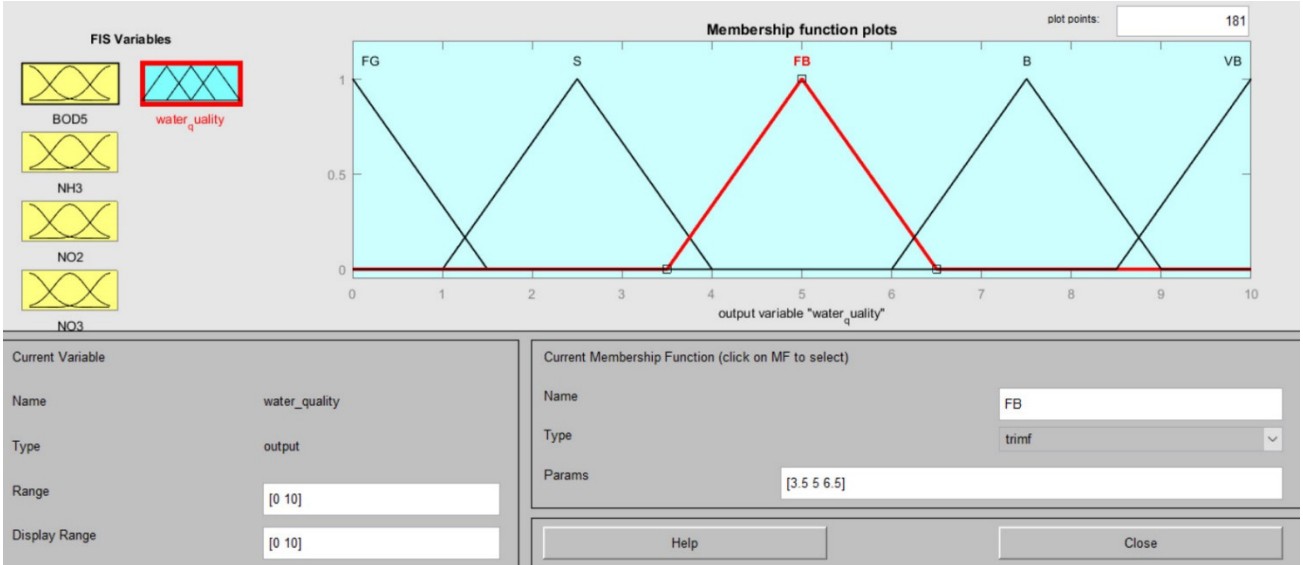

**Figure 6.** The five values of the output linguistic variable.

The final stage is defuzzification, when fuzzy linguistic output variables are converted into exact numbers.

Figure 7 shows an example of a defuzzification step. For example, for input variables (BOD$_5$, NH$_3$, NO$_2$ and NO$_3$), whose numerical values are (9; 2; 0.3; 60), respectively, the value of the output variable WQI is 5.03.

The output variable WQI was calculated for 977 monitoring records.

After applying the created fuzzy logic model, a matrix was formed that contained 977 rows and 5 columns (4 columns of input variables (BOD$_5$, NH$_3$, NO$_2$ and NO$_3$), and 1 column (WQI) of output data). Before using the data in the ANN model, they were normalized. The normalization of the input data meant the calculation of the arithmetic mean and standard deviation. Further, the arithmetic mean was subtracted from the input data, and the result was divided by the standard deviation. The database was split into a training set (70%), a validation set (15%) and a testing set (15%). The next stage of this research was the training, validation and testing of six ANN models. Table 3 shows the performance of six ANN models. The models were tested at 20, 50 and 100 epochs. With increasing epochs greater than 100, no improvement in model performance was observed.

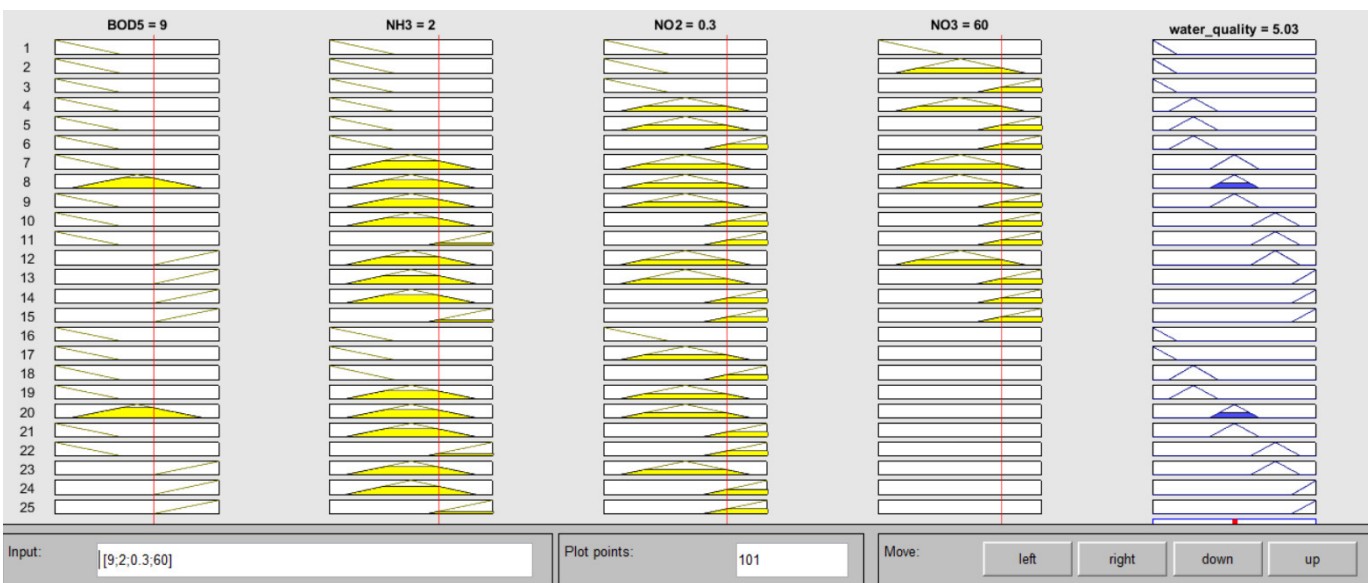

**Figure 7.** The example of defuzzification.

**Table 3.** Parameters of ANN models.

| Parameters | Models | | | | | | | | | | | | | | | | | |
|---|---|---|---|---|---|---|---|---|---|---|---|---|---|---|---|---|---|---|
| | **ANN 1** | | | **ANN 2** | | | **ANN 3** | | | **ANN 4** | | | **ANN 5** | | | **ANN 6** | | |
| **activator** | Sigmoid | | | Sigmoid | | | ReLU | | | ReLU | | | softmax | | | softmax | | |
| **optimizer** | Adam | | | SGD | | | Adam | | | SGD | | | Adam | | | SGD | | |
| **epochs** | 20 | 50 | 100 | 20 | 50 | 100 | 20 | 50 | 100 | 20 | 50 | 100 | 20 | 50 | 100 | 20 | 50 | 100 |
| $R^2$ | 0.872 | 0.919 | 0.944 | 0.830 | 0.896 | 0.937 | 0.396 | 0.553 | 0.348 | 0.355 | 0.302 | 0.398 | 0.898 | 0.933 | 0.964 | 0.833 | 0.911 | 0.916 |
| *MAPE, %* | 15.8 | 12.5 | 10.8 | 16.6 | 14.0 | 11.1 | 17.8 | 14.1 | 14.4 | 16.5 | 14.7 | 13.8 | 13.6 | 11.2 | 9.6 | 15.1 | 13.8 | 13.2 |

The comparison of six ANN models showed that the best performance results (lowest average absolute error in percentage and the largest coefficient of determination) were in ANN 5. In this network, the Adam loss function optimizer and the softmax activation function were used. The optimal *MAPE* and $R^2$ values were reached at the 100th epoch and were equal to 9.6% and 0.964, respectively.

The following description of the results of the study refers to the ANN 5 network and allows us to evaluate its adequacy and the possibility of using it to predict the output value in new datasets. Figure 8 shows a comparison of the loss function for the training and testing sets.

Figure 9 shows a comparison of the quality metric for training and testing sets.

Table 4 shows the loss function (*MSE*) and quality metric (*MAE*) values of the training and validation dataset for the initial and final epochs.

**Table 4.** Loss function (*MSE*) and quality metric (*MAE*) values of the training and validation set.

| Epochs | Training Set | | Validation Set | |
|---|---|---|---|---|
| | *MSE* | *MAE* | *MSE* | *MAE* |
| **1** | 7.2669 | 0.9170 | 5.3289 | 0.4579 |
| **2** | 3.9745 | 0.2824 | 3.0942 | 0.1824 |
| **3** | 2.4992 | 0.1372 | 2.2143 | 0.1204 |
| **4** | 1.8949 | 0.1013 | 1.8098 | 0.1000 |
| **5** | 1.5088 | 0.0817 | 1.4278 | 0.0788 |

**Table 4.** *Cont.*

| Epochs | Training Set | | Validation Set | |
|---|---|---|---|---|
| | *MSE* | *MAE* | *MSE* | *MAE* |
| . . . | . . . | . . . | . . . | . . . |
| **94** | 0.1087 | 0.0106 | *0.1494* | *0.0138* |
| **95** | 0.1088 | 0.0106 | 0.1512 | 0.0140 |
| **96** | 0.1086 | 0.0106 | 0.1501 | 0.0140 |
| **97** | 0.1075 | 0.0105 | 0.1545 | 0.0144 |
| **98** | 0.1083 | 0.0107 | 0.1530 | 0.0140 |
| **99** | 0.1064 | 0.0105 | 0.1541 | 0.0136 |
| **100** | *0.0961* | *0.0104* | 0.1548 | 0.0146 |

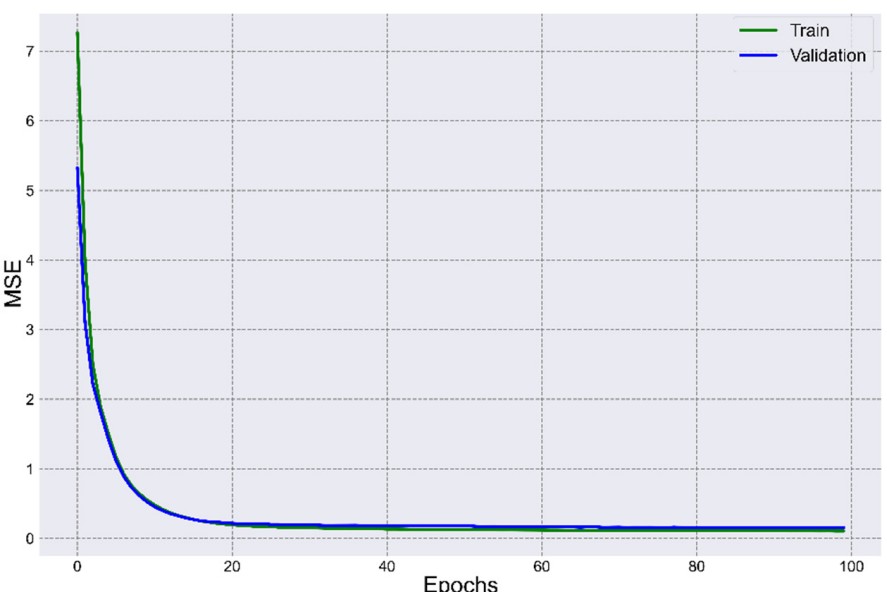

**Figure 8.** Comparison of MSE loss function for training and validation set.

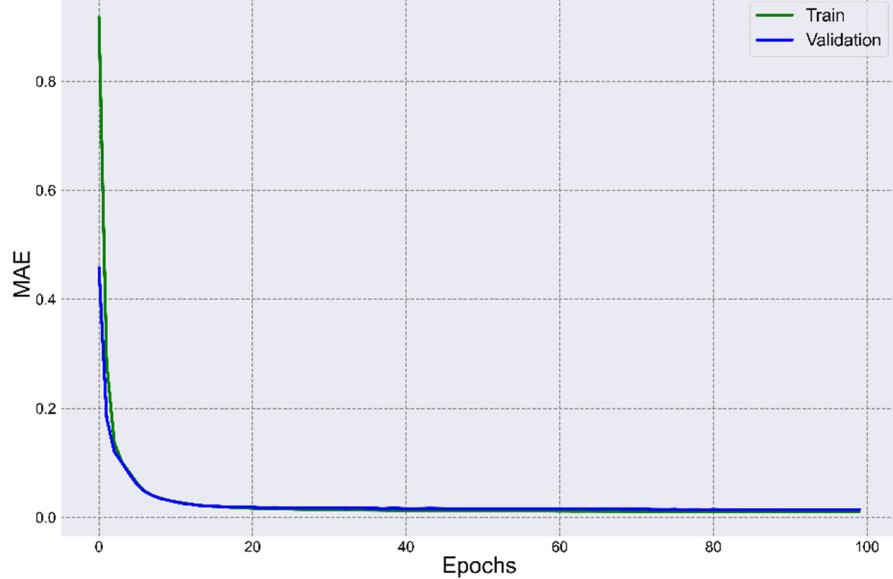

**Figure 9.** Comparison of the quality metric for training and testing sets.

For the training set, the value of the *MSE* loss function at the 1st epoch was 7.2669 and decreased to 0.0961 at the 100th epoch. The value of the *MAE* metric was 0.9170 and decreased to 0.0104 at the 100th epoch. The lowest *MSE* and *MAE* values were recorded at the 100th epoch. For the validation set, the value of the *MSE* loss function at the 1st epoch was 5.3289 and decreased to 0.1548 at the 100th epoch. The value of the *MAE* metric was equal to 0.4579 and decreased to 0.0146. The lowest values of *MSE* and *MAE* were recorded at the 94th epoch—0.1494 and 0.0138, respectively. In general, it can be argued that the values of the loss function (*MSE*) and the quality metric (*MAE*) of the training and validation sets did not differ much, and the ANN model was able to predict the WQI values with high accuracy.

Previously, in various scientific studies of water quality, scientists used the integration of several methods. Elkiran et al. [66] and Naja et al. [67] showed the possibility of using the ANFIS method to predict the quality of river water. The model proposed by the authors was able to overcome the shortcomings of ANN models (overfitting and falling into local minima) and systematized fuzzy logic with ANN, which allows problems that have uncertainty to be solved.

Huang et al. [68] synthesized a model that combined neural networks, fuzzy logic, wavelet transforms and GA and was capable of solving non-linear problems and coping with data oscillation problems. Hu et al. [69] used a Long Short-Term Memory (LSTM) neural network. The results showed a prediction accuracy of 98.97%, which confirms the possibility of using the LSTM model for long-term prediction. Ding et al. [70] collected 23 water quality parameters and, using raw data compression methods, moved to 15 aggregate indices. Then, a genetic algorithm approach was applied to optimize the BPNN parameters, which made it possible to achieve an average forecast accuracy of 88%.

In general, it can be argued that the use of a combination of mathematical modeling methods to assess water quality was successful. One of the main methods for analyzing an ANN model is the comparison of various indicators (mean absolute percentage error, MAPE, root mean square error, RMSE and root mean square error, *MSE*), as well as coefficients of determination ($R^2$) and regression (*R*).

Yesilnacar et al. [71] predicted groundwater nitrate concentrations using four ANN inputs. The model tracked the experimental data very accurately, with an *R* value of 0.93. Antanasijevic et al. [72] analyzed the performance of Regression Neural Network (RNN), GRNN and MLP models for water quality prediction. The results showed that the error in the RNN model on the test data was less than 10%, and the GRNN error was lower than that of MLP. Alqahtani et al. [73] compared various machine learning models, such as gene expression programming (GEP) and ANNs, with a random forest (RF) model, for river water quality prediction. The neural network showed an $R^2$ value of 0.88 and an *MSE* of 5.5% for the validation dataset. Akiner and Akiner [74] used an ANN model to estimate the concentration of dissolved oxygen in water. The model performed better ($R^2 = 0.856$ and *MSE* = 9.5%) than the traditional multiple linear regression analysis. Shah et al. [75] analyzed the effectiveness of using GEP, ANN and linear regression model to predict the level of dissolved solids and electrical conductivity. The outcomes were evaluated using various performance measures, error levels and external criteria. The artificial neural network showed an $R^2$ value of 0.90 and an *MSE* of 13.1% for the test dataset. Kulisz et al. [76] used neural network models to predict groundwater quality in industrial areas. The best parameters were obtained for a network with five input neurons (conductivity, pH and calcium, magnesium and sodium ions), in addition to five hidden layer neurons. The authors summarized that ANNs demonstrated the ability to predict the water quality index with the desired level of accuracy (*RMSE* = 65.13%; $R^2 = 0.9984$).

Thus, comparing the *MAPE* and $R^2$ indicators of the created ANN model (*MAPE* = 9.6%; $R^2 = 0.964$) with other models, we can state that, in general, the level of agreement between the predicted and target data is satisfactory.

## 5. Conclusions

The object of the study was to monitor data on the quality of water of the Western Bug River (Ukraine) for the period 2016–2021. On the basis of the conducted studies, we proposed a modification of the method for assessing the WQI, which is used in Ukraine. The change consists of the division of chemical indicators into three groups, depending on the level of negative impact on the quality of surface waters. Based on the grouping of chemical indicators, it is proposed to use an algorithm for assessing water quality, which consists of two stages. The next step of the study was the modeling of the WQI assessment using fuzzy logic. The model used input variables ($BOD_5$, $NH_3$, $NO_2$ and $NO_3$) and calculated the WQI as an output value. The final stage of the study was to create an artificial neural network model for WQI prediction. We tested six ANN models using different activation functions and loss function optimizers.

The network with the Adam loss function and the softmax activation function showed the best performance results. The optimal values of *MAPE* and $R^2$ were reached at the 100th epoch and were equal to 9.6% and 0.964, respectively. The comparison of the *MAPE* and $R^2$ indicators of the created ANN model with other models for assessing water quality showed that the level of agreement between the forecast and target data was satisfactory. Future studies will concern the interpretation of the proposed model in the assessment of groundwater quality for the study area. This will create a database for the comparison of groundwater and surface water quality in the same area.

**Author Contributions:** Conceptualization, R.T., Y.T., A.K., A.M., M.L.-S. and K.R.; methodology, Y.T. and R.T.; software, R.T.; validation, Y.T., A.K. and R.T.; formal analysis, Y.T., R.T., A.K., M.L.-S., A.M. and K.R.; investigation, Y.T. and R.T.; resources, Y.T. and R.T.; data curation, A.K., A.M., M.L.-S., K.R. and Y.T.; writing—original draft preparation, A.M., K.R. and R.T.; writing—review and editing, A.K., M.L.-S. and A.M.; visualization, Y.T. and R.T.; supervision, R.T.; funding acquisition M.L.-S., R.T., Y.T., A.K. and A.M. All authors have read and agreed to the published version of the manuscript.

**Funding:** This research received no external funding.

**Institutional Review Board Statement:** Not applicable.

**Informed Consent Statement:** Not applicable.

**Data Availability Statement:** The datasets generated and analyzed during the current study are available from the authors upon reasonable request.

**Conflicts of Interest:** The authors declare no conflict of interest.

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
