# Peer review of "A Study of Assessment and Prediction of Water Quality Index Using Fuzzy Logic and ANN Models"

_sustainability, doi:10.3390/su14095656_

Round 1

Reviewer 1 Report

The article deals with developing algorithm for assessing water quality of the Western Bug River (Ukraine) using fuzzy logic and ANN.

Overall the article is very good and I think it should be published in quality journal "Sustainability"

Only modification needed in figure 2, authors are advised to represent neurons with dots (....) if they cannot show all in hidden layers so that the readers get clear idea 

Author Response

The reviewer's answer is in the attached file.

Reviewer 2 Report

In the presented study, fuzzy logic and artificial neural network are used to assess the state of water quality in in the Western Bug River (Ukraine). It is commonly known that pollution poses a substantial threat to surface water bodies all over the world. Regular monitoring and quality analysis utilizing effective approaches are therefore required. In that regard, the offered topic may be of interest to journal readers, but the work will need to be revised before it can be examined further. The following are some of the concerns that must be addressed in the manuscript:

“Abstract”

  • In the abstract, it is essential to highlight the material merits and work novelty. The summation of the results has not been well presented.
  • L16-17: the statement, “the modeling of the WQI assessment was using Fuzzy Logic.” Is redundant, the authors should paraphrase it.
  • Also, the abstract contains a lot of abbreviations, the authors should consider removing them.

“Introduction” part:

  • This part should be revised with more focus in coherence, the flow of ideas from paragraph to another should well arranged. It should also be noted that, fuzzy logic and ANN models have widely used in the field of water quality monitoring. Therefore, the authors should focus on highlighting the novelty of the current work in the introduction section.
  • The general formatting of the manuscript is also poor, the authors should check that.

“Materials and methods”

  • In this section, the authors should include the longitude and latitude of the case study.
  • There is little problem with the methodologies, The approaches are fine; the authors spent more time presenting the fundamentals of the models than what they did themselves. To assist the reader, the authors should thoroughly clarify the procedures used during the study step by step.
  • L194: The authors mentioned “Mandani” as a fuzzy inference, they should correct it to “Mamdani”.

“Results”

  • In think there is some information in the results section that was supposed to be either in the materials and methods or introduction section. For instance, Table 1 should be in the data collection section and water quality characteristics in the materials and methods section.
  • In addition, the authors stated maximum contamination values in Table 1, however it is unclear which standards they were based on. The authors should emphasize the authority that established the recommendations, and a reference would be helpful.
  • The author should add further interpretation and explanation of the work and results obtained. More details and explanations regarding the figures and results obtained should be added to help the reader.

“Discussion”

The discussion section is superficial, more discussion of their own results, as well as setting them in comparison to those discovered in the literature, is needed in the discussion section. Alternatively, I would suggest merging the results and discussion sections.

“Conclusion”

Firstly, the conclusion should be merged into one paragraph, then should be revised based on the material, the objective, the main results obtained, as well as clarification of the future applicability of the study.

Language and writing style should also be carefully checked in the manuscript.

Author Response

(The authors gave the same response as above.)

Reviewer 3 Report

Comments from the reviewer:

  1. Introduction:

Introduction to be rewritten in three separate paragraphs for reasons of review, literature review and research purposes. The structure of the paper should be rewritten in 3 separate paragraphs. In the first paragraph, information about the subject under study, in the second paragraph, the study of different researchers and the studied indicators, and in the third paragraph, the objectives of the research should be stated.

1-1) Lines 54-55: You can add the newer references. For example:

  1. A) Ali Rezaei, Hossein Hassani, Sara Hassani, Nima Jabbari, Seyedeh Belgheys Fard Mousavi, Samira Rezaei. Evaluation of groundwater quality and heavy metal pollution indices in Bazman basin, southeastern Iran. 2019, Groundwater for Sustainable Development.

  1. B) Ali Rezaei, Hossein Hassani, Mohammad Hayati, Nima Jabbari, Rahim Barzegar. Risk assessment and ranking of heavy metals concentration in Iran’s Rayen groundwater basin using linear assignment method, 2017, Stochastic Environmental Research and Risk Assessment, 1436-3240

1-2) Line 62-84, 95-101: Please transfer to material and methods section.

2) The study area comes in a separate section before materials and methods section.

3) Figures 3, 4, 5 and 6 should be presented more clearly.

4) Mention the reason for selecting the collected samples for analysis. What are the principles of points sampling?

5) Before modeling, the basic statistics section of the parameters involved in the model should be added.

6) How much is the validity of the modeling and can it be trusted? State the reasons for the verification.

 7) In my opinion, this model can be better interpreted for groundwater. If you have information about the groundwater of the study area, compare it with the surface information and model it based on the obtained results.

8) The chart in Figure 3 is not clear. Please correct it.

Author Response

(The authors gave the same response as above.)

Reviewer 4 Report

Reviewer comments

Manuscript title: “A Study of assessment and prediction of water quality index using Fuzzy Logic and ANN Models " is an interesting article related to the assessment and prediction of water quality index using Fuzzy Logic and ANN Models.

The results are interesting. However, most of the figures need to be improved and updated.

Minor comments:

  1. In the introduction, recent literature needs to update. The author should add current studies that related to the present study
  2. Please write the full form of the ANN model
  3. Line 17: BOD5, NH3, NO2, NO3……correct with BOD5 , NH3, NO2, NO3
  4. Line 299: O2 correct with….O2
  5. Line 304: PO43--……..Correct with PO43-
  6. Figure 3: The authors should correct and improve the Figure 3 some of the text is not visible properly
  7. Please improve quality of the figures 4 & 5, If possible please add a caption note that describes the data interpretation.
  8. Authors should discuss more, present results, with the latest literature related to the present study
  9. The full Manuscript needs to be read and corrected carefully to reduce typological errors

Author Response

(The authors gave the same response as above.)

Round 2

Reviewer 2 Report

The authors addressed most of the issues highlighted in the previous version and improved the manuscript. However, some minor issues still need to be addressed:

  • I suggest removing sub-sections in the introduction. The flow of ideas can be well arranged without dividing the introduction into sub-sections.
  • The previous comments raised the issue of latitude and longitude, but the authors did not work on that.
  • The resolution of the figures should be improved; the text in figures 4 to 7 is not visible.
  • Again, the writing and language style should be well checked in the manuscript.

Author Response

The answer to the reviewer is in the attached file.
